
2

# Soil Conservation in the 21st Century: Why we need Smart Agricultural Intensification

Gerard Govers[1], Roel Merckx[1], Bas van Wesemael[2], Kristof Van Oost[2]

[1] KU Leuven, Department of Earth and Environmental Sciences, Celestijnenlaan 200E, 3001 Leuven, Belgium
[2] Université Catholique de Louvain, Earth and Life Insititute, 3 Place Louis Pasteur, B-1348 Louvain-la-Neuve, Belgium

*Correspondence to:* Gerard Govers (gerard.govers@kuleuven.be)

**Abstract.** Soil erosion severely threatens the soil resource and the sustainability of agriculture. After decades of research this problem persists, despite the fact that adequate technical solutions now exist for most situations. This begs the question as to why soil conservation is not more rapidly and more generally implemented. Studies show that the implementation of soil conservation measures depends on a multitude of factors but it is also clear that rapid change in agricultural systems only happens when a clear economic incentive is present for the farmer. Conservation measures are often more or less cost-neutral which explains why they are often less generally adopted than expected. This needs to be accounted for when developing a strategy on how we may achieve effective soil conservation in the Global South, where agriculture will fundamentally change in the next century. In this paper we argue that smart intensification is a necessary component of such a strategy. Smart intensification will not only allow to make soil conservation more economical, but will also allow to make significant gains in term of soil organic carbon storage, water efficiency and biodiversity, while at the same time lowering the overall erosion risk. While smart intensification as such will not lead to adequate soil conservation, it will facilitate it and, at the same time, allow to offer the farmers of the Global South a more viable future.

**Introduction**

The terrestrial land surface provides critical services to humanity and this is largely possible because soils are present. Humanity uses ca. 15 million km² of the total Earth's surface as arable farmland (Ramankutty et al., 2008). Besides this, ca. 30 million km² is being used as grazing lands: on all these lands grow plants which are either directly (as food) or indirectly (as feed, fibre or fuel) used by humans for nutrition and a large range of economic activities. Agricultural areas, especially areas used as arable land, have often been selected because they have soils that make them suitable for agriculture. But it is not only the soils on agricultural land that provide humanity with essential services. Also on non-agricultural land soils provide the necessary rooting space for plants, store the water necessary for their growth and provide nutrients in forms that plants can access. Both on agricultural and non-agricultural land soils are host to an important fauna whose diversity is, by some measures, larger than that of its aboveground counterpart (De Deyn and Van der Putten, 2005). Both on agricultural and non-agricultural land soils store massive amounts of organic carbon, the total amount of which (ca. 2500 Gt, Batjes, 1996; Hiederer and Köchyl, 2012) is much larger than the amount of carbon present in the atmosphere (ca. 800 Gt). Importantly, organic carbon storage per unit area is generally much higher on non-agricultural land (Poeplau et al., 2011; Hiederer and Köchyl, 2012). By allowing plants to grow, soils significantly contribute to the terrestrial carbon sink, which removes an amount equal to 30-40% of the carbon annually emitted by humans from the atmosphere (Le Quere et al., 2009). Soils, both those on agricultural and non-agricultural lands, are therefore a vital part of humanity's global life support system, just like the atmosphere and the oceans. An Earth without soils would be fundamentally different from the Earth as we know it and would, in all likelihood, not be able to support human life as we know it.

No further arguments should be necessary to protect soils from the different threats posed to them by modern agriculture and other human activities. Yet, as is the case with many other natural resources, soils are under intensive pressure. Organic carbon loss, salinization, compaction and sealing all threaten the functioning of soils to different extents in different areas of the world. One of the most important and perhaps the ultimate threat posed to soils is accelerated erosion due to agricultural disturbance. When soils are used for farming their natural vegetation cover is removed and they are often disturbed by tillage. The result is that, under conventional tillage, erosion rates by water on arable land are, on average, up to two orders of magnitude higher than those observed under natural vegetation. This acceleration creates a major imbalance as soil production is outstripped by soil erosion by a factor 10-100 so that soil is effectively mined (Johnson, 1987; Montgomery, 2007; Vanacker et al., 2007b). Eroded soil is, in many cases, truly lost and cannot be restored (although there are exceptions to this rule), which explains why land prices in areas heavily affected by erosion may remain lower than expected, even when excessive erosion has been halted for several decades (Hornbeck, 2012).

It is rather surprising that agricultural soil erosion still is such an important problem. Pre-industrial societies such as the Inca already understood that erosion threatened agricultural productivity and used soil conservation techniques such as terracing for centuries (Krajick, 1998). In France, environmental degradation by excessive water erosion of mountain hillslopes literally ruined the livelihood of entire mountain communities at the end of the 19th century (Robb, 2008). A similar situation developed in Iceland where excessive wind and water erosion forced entire villages to be abandoned in the same period. In both countries overexploitation of the natural environment by subsistence farmers through excessive deforestation and overgrazing were key factors. Both countries responded to this situation: in Iceland the first soil conservation service of the world was founded in

1907 (Arnalds, 2005), while France started an extensive programme to restore its mountain environments (RTM)
as early as 1860 (Lilin, 1986). In the United States, the Dust Bowl years (1930s) moved the erosion problem high
up the political agenda: President Franklin Roosevelt not only erected a Soil Conservation Service but also,
famously, said 'A nation that destroys its soils destroys itself' (FAO and ITPS, 2015).
One might therefore expect that, by now, detailed information would exist on the status of the global soil resource
and the necessary measures would have been taken to stop soil degradation due to human action and/or mitigate
the consequences. Yet, this is clearly not the case: recent estimates of human-induced agricultural erosion amount
to 25-40 Gt yr$^{-1}$ for water erosion, ca. 5 Gt yr$^{-1}$ for tillage erosion and 2-3 Gt yr$^{-1}$ for wind erosion (Van Oost et al.,
2007; Govers et al., 2014). Measured soil production rates are, on average, ca. 0.036±0.04 mm yr$^{-1}$ (Montgomery,
2007) and are even lower on most agricultural soils because agricultural soils have a certain thickness and soil
production rates decrease with increasing soil depth (Stockmann et al., 2014). Thus, over all agricultural land
(arable and pasture) total soil formation would amount to maximum ca. 2 Gt yr$^{-1}$ which implies that the global soil
reservoir is depleted by erosion at a rate which is ca. 20 times higher than the supply rate.  Although these numbers
are only an approximation (for instance, they do not account for the fact that eroded soil may be re-deposited on
agricultural land) they clearly illustrate that we are still far away from a sustainable situation: the rate at which the
soil resource is being  depleted is, over the longer term, a clear threat to agricultural productivity (FAO and ITPS,
2015). The loss of mineral soil is not the only issue: soil erosion also mobilises 23-42 Tg yr$^{-1}$ of nitrogen and 14-
26 Tg yr$^{-1}$ of phosphorus (Quinton et al., 2010). These numbers may be compared with the annual application rate
of mineral fertilizers, which are ca. 122 Tg yr$^{-1}$ for N and ca. 18 Tg yr$^{-1}$ of mineral P respectively. At 2013 USA
mineral fertilizer prices of ca. 1.35 USD (kg N)$^{-1}$ and ca. 4.75 USD (kg P)$^{-1}$, (http://www.ers.usda.gov/data-
products/fertilizer-use-and-price.aspx) the annual amount of fertilizers mobilised by soil erosion is equivalent to
ca. 35 billion US $ for N and ca. 80 billion US $ for P: this is a significant financial loss, even if one considers that
the total global agricultural food production is nowadays valued at ca. 4000 billion US $
(http://faostat.fao.org/site/613/DesktopDefault.aspx?PageID=613#ancor). Most of these soil and nutrient losses
take place in the hilly and mountain areas in the so-called Global South: a recent scientific appraisal by FAO and
the ITPS (the Intergovernmental Technical Panel on Soils) showed that erosion problems are still increasing in
Africa, Latin-America and Asia (FAO and ITPS, 2015). The situation is perceived to be improving in Europe and
North America (FAO and ITPS, 2015), albeit that also in these regions soil losses are often still above the tolerable
level (Verheijen et al., 2009). Thus, it is especially the agriculture in the Global South (Latin America, Africa, the
developing nations of Asia and the Middle East), where it is often one of the main economic activities, which
suffers excessively from these losses.
In this paper we reflect on why, despite these clear facts, effective soil conservation is not yet a done deal and what
might be done about this. We argue that there is a need for a novel vision on soil conservation in the Global South,
shifting the focus away not only from the technical issues of soil conservation but also away from soil conservation
as such. Soil conservation efforts need to be framed into a general vision on how agriculture will develop in the
South: this vision needs to account for soil protection, but must also guarantee food security and allow the
development of an agricultural system that does provide a sufficient income to farmers. We will first assess
possible reasons as to why soils do not yet get the protection they deserve. Thereafter we will discuss the building
blocks of a vision on future soil conservation.
**The status of soil conservation**
**Do we have the necessary data to guide soil conservation?**
Investing in the application of soil conservation measures is only meaningful when erosion rates are higher than
acceptable. This can most easily be established when erosion rates can reliably be quantified. Quantitative
information is indeed available for North America and Europe (Cerdan et al., 2010; NRCS, 2010). However, the
quality of our estimates of soil erosion rates by water for other areas on the globe is often poor. Sometimes,
estimates are based on a limited number of data which are simply extrapolated to larger areas: this often leads to
bias, simply because erosion rates are generally measured at locations where erosion intensity is much higher than
average (Boardman, 1998; Cerdan et al., 2010). Also when models are used to make an extrapolation, estimates
are often incorrect. This is due to two reasons: (i) the models that are used are often improperly calibrated, i.e.
model parameters are set to values that are not appropriate for the location under consideration and (ii) the model
parameterization may be correct but the spatial data used to drive the model are inappropriate. A typical example
of the latter is when slope lengths are directly derived from a DTM so that the impact of slope breaks such as field
borders is not accounted for (e.g. Yang et al., 2003). This can lead to a considerable overestimation of erosion
rates (Desmet and Govers, 1996; Cerdan et al., 2010; Quinton et al., 2010). Erroneous predictions do not only
make it difficult to identify the most vulnerable areas in which conservation measures are most urgent: they may
also invalidate the cost-benefit evaluations of soil conservation programs and lead to disinformation of the general
public about the extent and severity of the problem.
**Although there is a clear need for better, quantitative data on erosion rates, the lack of such data is not the most**
**important explanation as to why excessive soil erosion often still goes unchecked. While it may indeed be difficult to**
**quantify erosion rates correctly, it is much easier to identify those areas where intense soil erosion is indeed a problem**
**and where action is necessary, whatever the exact erosion rates are. This is, after all, what institutions such as the soil**
**conservation services of Iceland and the United States did long before accurate erosion measurements were available.**
**Simple visual observations on the presence of rills and gullies or wind deflation areas are clear indications that the**
**implementation of conservation measures is necessary (**

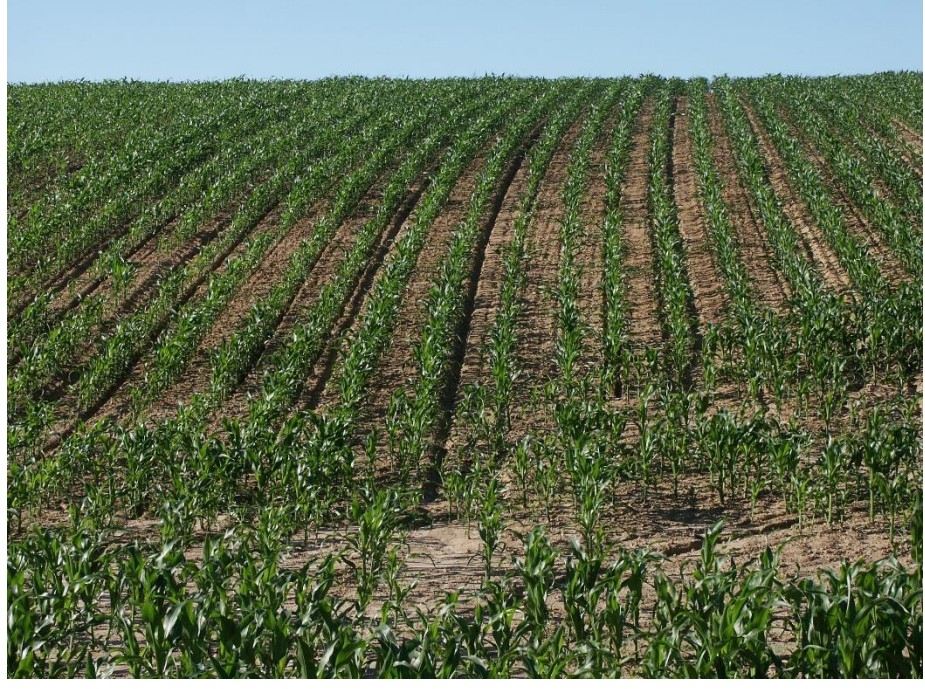


Figure 1). Another reason why an exact quantification is not always necessary is that conservation measures
generally are not proportional: Their implementation is most often of a yes/no type: one can decide whether or not
to implement conservation tillage, but not by how much.
**Do we have the necessary technology for soil conservation?**
There is no doubt that soil conservation technology has matured over the last decades: we now have the tools to
effectively reduce erosion rates to acceptable levels in many, if not all, agricultural systems. Conservation tillage
is the tool of choice in many areas, especially in the Americas. This is hardly surprising: erosion plot research has
consistently shown that water erosion rates under conservation tillage are reduced by one to two orders of
magnitude in comparison to conventional systems (Montgomery, 2007; Leys et al., 2010). Moreover, the
effectiveness of conservation tillage as calculated by plot studies is likely to be underestimated: for various reasons
the effectiveness of conservation does increase if the slope length increases (Leys et al., 2010). As a consequence,
water erosion rates under conservation tillage on moderate slopes are generally very low (< 1 t ha y) and often
comparable to those occurring under natural vegetation (Montgomery, 2007). Conservation tillage may also be
used to drastically control wind erosion not only because residue cover does reduce the shear stress to which soil
particles are exposed but also because the presence of residue helps to keep the surface soil layer moist, thereby
increasing its shear strength.
Conservation tillage is not always the best tool. It may be difficult or impossible to apply with certain crops, such
as potatoes grown on ridges, and/or difficult to introduce into specific agricultural systems as it may affect the
overall workload or the gender balance of the workload (Giller et al., 2009). It may also not be sufficient to
implement conservation tillage as processes such as gully erosion may not be effectively controlled and may in
some cases even be enhanced by conservation tillage as the latter is much more effective in reducing erosion than
in reducing surface runoff (Leys et al., 2010). However, also in such cases technological solutions do exist: they
can consist of infrastructural measures such as stone bunds and terrace building in combination or  vegetation
measures such as grassed waterways, but also proper land use allocation can make a significant difference. Water
and wind erosion rates can often be reduced to acceptable levels through the use of such measures in combination
with modifications of tillage techniques and crop rotations (Sterk, 2003; Valentin et al., 2008; Nyssen et al., 2009).
Not only arable land can be affected by excessive erosion. Grazing lands may suffer from a drastic reduction in
vegetation cover due to overgrazing and compaction, again resulting in excessive water and/or wind erosion with
rates up to two orders of magnitude higher than those observed under natural conditions (Vanacker et al., 2007b).
Reduction of grazing pressure (at least in a first stage) and the introduction of controlled grazing are key strategies
(i) to restore the vegetation cover and (ii) to allow these lands to become productive again so that they can be
sustainably used (Mekuria et al., 2007). Such measures can be further supported by the planting of trees (Sendzimir
et al., 2011). Reforestation may also be a solution as it reduces erosion rates to near-natural levels but it has evident
implications for the type of agriculture that can be supported (Vanacker et al., 2007b). Thus, as is the case on
arable land, the key to erosion reduction on grasslands is in most cases the maintenance or restoration of a good
vegetation cover, possibly supported by technical measures.
Erosion in agricultural areas is often not directly related to agricultural activities but also to the infrastructure
related to these activities such as roads and field boundaries. Unpaved roads on sloping surfaces are not only
important sources of sediment in many agricultural areas (Rijsdijk et al., 2007; Vanacker et al., 2007a) but also in
cities (Imwangana et al., 2015). Water is often concentrated at field boundaries therebyleading to gully formation
(Poesen et al., 2003). Again, the necessary technological know-how to control such erosion phenomena is
available: check dams, better water drainage infrastructure, the implementation of field buffer zones and a better
landscape organisation all help to reduce sediment production on road networks and in built-up areas.
**Why then is soil conservation not more generally adopted?**
Thus, neither the lack of conservation technology nor the lack of data on the erosion hazard can fully explain why
efficient soil conservation measures are still not implemented on most agricultural land, especially in the Global
South. It has indeed long been clear that several factors other than (the lack of) scientific knowledge or data hamper
the adoption of conservation tillage. These factors include the training level of the farmer, the farm size and work
organisation as well as access to information.  However, a thorough analysis by Knowler and Bradshaw (2007)
showed that the effect of these variables was often ambiguous (when different studies are compared) and that few,
if any, variables showed a consistent effect.  One might conclude from this that changing farming practices must
be inherently difficult, as our understanding of controlling factors is relatively poor and many barriers to the
adoption of novel technology need to be overcome. This is not only a problem in the Global South: also in Europe
the adoption of conservation tillage is slow in many countries due to a multitude of factors, including the fact that
soil tillage is deeply rooted in the culture of many farmers (Lahmar, 2010).
Clearly, farming systems are, to some extent, 'locked in': they rely on well-tried technology, division of labour
and crop types and are therefore difficult to change. There are, nevertheless, also cases where farming systems
change rapidly and conservation technology is quickly adopted. Once the necessary technology was available,
conservation tillage spread very rapidly through most of Argentina and Brazil: in Argentina, it took ca. 20 years
(from 1990 to 2010) to bring ca. 80% of the arable land under no-till (Peiretti and Dumanski, 2014), thereby
effectively halting excessive soil erosion on most of the arable land of the country. In Brazil, more than 25 million
ha of land was under no-tillage in 2006, whereas the technique was virtually unused before 1990 (Derpsch et al.,
2010). Rapid changes in agricultural systems are not limited to the adoption of conservation tillage. When
subsistence farmers in remote areas gain access to profitable markets, very rapid changes can occur, even in areas
where existing technology is poor: such changes can have very negative effects in terms of soil degradation rates
as a switch to cash cropping may introduce crops to which a much higher erosion risk is associated (Valentin et
al., 2008). Thus, while cultural and technological barriers to change certainly do exist, farmers are most certainly
capable of rapid change. Whether such rapid change occurs critically depends on whether farmers think change
will bring them a personal gain.
This is where the problem lies. Under some conditions, the adoption of conservation technology is indeed clearly
economically beneficial to the farmer: this appears to be true for large farming operations in (sub-) tropical regions
growing cash crops such as soy beans (Peiretti and Dumanski, 2014). But in most other cases the direct benefits
of the implementation of conservation agriculture and/or other soil conservation measures are small, if they exist
at all. This appears to be the case for both large-scale mechanised agriculture in the temperate zone as well as for
marginal hillslope farming in developing countries (Knowler et al., 2001).  In both scenarios, potential savings are
offset by additional costs: in mechanized systems the cost of machinery and agrochemicals offsets savings in fuel
costs (Zentner et al., 1996; Janosky et al., 2002) while in traditional hillslope farming extra work hours are needed
to maintain conservation structures and some land has to be sacrificed to implement these structures, thereby
reducing overall yields (Nyssen et al., 2007; Quang et al., 2014). Importantly and contrary to common belief, crop
yields do not rise significantly in conservation systems if no additional inputs are provided: this is true for advanced
technological systems (Van den Putte et al., 2010; Pittelkow et al., 2015) as well as for tropical smallholder farming
(Brouder and Gomez-Macpherson, 2014). As a consequence, farmers often do not have direct incentives to
implement soil conservation measures and change becomes difficult to implement.
One may argue that benefits should not only be considered at the level of the individual farmer, but also at the
societal level, where soil conservation may generate co-benefits. Often carbon storage and biodiversity protection
under conservation systems are mentioned as important ecosystem services for which farmers could be paid.
Research in the last decade has consistently shown that carbon storage gains in conservation systems are lower
than was anticipated two decades ago and is generally well below 1 t C ha$^{-1}$ yr$^{-1}$ ((Oorts et al., 2007; Angers and
Eriksen-Hamel, 2008; Christopher et al., 2009; Eagle et al., 2012; Govers et al., 2013). Furthermore, paying
farmers to store carbon would only be viable at much higher carbon prices than the current market prices, which
are around 10-15 USD ton$^{-1}$ (Grace et al., 2012; Govers et al., 2013). Paying farmers at current market prices can
only generate a relatively small economic benefit for the farmer and prices would have to rise significantly for soil
carbon storage to become an important element on the farmers' balance sheet. On the other hand, soil conservation
generally has a positive impact on (soil) biodiversity on the farm land as soils are less frequently disturbed (Mader
et al., 2002; Verbruggen et al., 2010). Where agriculture is interspersed with densely populated areas, additional
co-benefits may consist of a reduction of flooding and/or siltation of sewage systems and water treatment plants,
which are important problems in many areas in Europe (Boardman et al., 1994). These benefits, however, are
difficult to convert to financial income for the farmer. This is not only because the economic value of increased
biodiversity on farmland is difficult to quantify but also because such on-farm benefits in biodiversity have to be
weighed against possible off-farm losses (see below). The reduction in flooding risk, on the other hand, will
generally not be considered as a benefit by society but rather as damage repair: the problems were caused by
agriculture in the first place.
**The way forward**
How then should we proceed to stimulate a more rapid adoption of soil conservation measures to protect the
world's soil resource? The answer to this question will obviously depend on the characteristics of the local agro-
ecological system. Agricultural systems show a large variety so that not only the factors impeding the adoption of
conservation tillage vary locally (Knowler and Bradshaw, 2007) but also the tools that societies have at their
disposal to reduce it.
Western societies with highly developed information systems tackle the problem by a policy combining regulation
(e.g. by forbidding the cultivation of certain crops on land that is very erosion-prone) and subsidies or
compensations in combination with well-guided campaigns to inform farmers on the potential benefits and risks
for themselves as well as for the broader society. Such combined approaches do have demonstrable success in
various parts of Europe and North America where farmers are not only well trained and highly specialized but also
depend to a large extent on subsidies, giving the administrations the necessary financial leverage to stimulate or
even coerce farmers (Napier et al., 1990). As a result erosion rates in North America have gone down considerably
over the last decades and are still declining (Kok et al., 2009). One may therefore assume that in these societies
erosion rates can be reduced to tolerable levels provided that the necessary policies are maintained and/or

strengthened. Countries having a strong central government that can impose decisions on land use and soil conservation, as is the case in China, can successfully reduce erosion: the excessive erosion rates on the Chinese Loess Plateau were strongly reduced through massive government programs implementing erosion control measures (Chen et al., 2007; Zhao et al., 2016)

These approaches are, at present, not possible in most countries of the Global South. Many governments in the Global South are not able to implement a successful soil conservation policy as they do not dispose of the necessary data and/or the necessary political and societal instruments to do so. At first sight it may therefore appear unlikely that soils will become effectively protected in most of the developing world within a foreseeable time span. Yet this conclusion foregoes the fact that agriculture in the Global South, and especially in sub-Saharan Africa, will see fundamental changes in the next decades. At least three fundamental tendencies can be identified that will change the nature of agriculture in the Global South in the 21$^{st}$ century: these should be accounted for when developing a vision on soil conservation.

*In many areas where soils are most seriously threatened, the human population will continue to grow strongly.* In the next decades, the locus of world population growth will shift in an unprecedented manner. Population growth in the North has stopped and many regions in the Global South will follow suit in the next decades: Asia is expected to reach its maximum population around 2050. China's population will peak around 2030 and that of India no later than 2070. Latin America will follow around 2060 (http://esa.un.org/unpd/wpp/, Lutz and KC, 2010; Gerland et al., 2014). Sub-Saharan Africa is a different matter: here the demographic transition started only after the Second World War and the population will continue to grow rapidly during most of the 21$^{st}$ century. As a result of these diverging tendencies the distribution of the world's population will have changed beyond recognition in 2100: Europe's share in the global population will have fallen from its maximum of ca. 22 % in 1950 down to ca. 6 % in 2100, while the share of Africa will rise from ca. 9 % in 1950 to ca. 39 % in 2100 (http://esa.un.org/unpd/wpp/). *The population in the South will also become more urban.* By 2050 ca. 2/3 of the global population is expected to live in cities (as compared to ca. 55% at this moment). Urbanisation rates are especially high in Africa where the fraction of urban population is expected to increase from 40% in 2014 to 55% in 2050 and in Asia, where urbanisation will increase from ca. 47.5% to ca. 65% over the same period (United Nations, 2014). There is no alternative for this evolution: despite all their problems, cities are the engines of modern economic development as they allow a population to create the added value that is so desperately needed through advantages of scale, intense interaction and exchange (Glaeser, 2011). This is the fundamental reason of the attractiveness of cities and the major factor explaining rural to urban migration: poor rural populations perceive the city as a place of opportunity and moving there as an opportunity to improve their own lives or at least those of their children (Perlman, 2006; Saunders, 2011). A consequence of this massive migration movement is that rural populations rapidly age and that the average farm worker is significantly older than the average non-farm worker (40 vs. 34 years in Africa, http://www.gallup.com/poll/168593/one-five-african-adults-work-farms.aspx). Clearly the evolution sketched above is a generalisation: local dynamics depend, amongst others, on the presence of attractive labour opportunities in the cities and the local availability of land (Ellis-Jones and Sims, 1995).

It is not overly optimistic to expect that, while population growth continues, at the same time *these populations will gain in purchase power.* While incomes in southern Asia and especially sub-Saharan Africa are nowadays much smaller than those in the North, their growth rates are, fortunately, much bigger. For example, Ethiopia's economy has, over the last decade, consistently been growing at 8 to 10% per year, leading to a rise of the per

capita Gross National Income from 110 US $ (2015 dollars) in 2004 to 550 US $ in 2015
(http://data.worldbank.org/country/ethiopia).
Combined, these tendencies will lead to an increased market demand for food. Furthermore, diets will move away
from a diet largely based on cereals towards a more varied (but not necessarily healthier) food palate in which
meat is likely to have a larger share than is currently the case. Global estimates therefore sometimes predict that
global food production (in terms of kcal) will increase more or less double in the first half of the 21$^{st}$ century
(Tilman et al., 2011) but an increase in demand by 60-70% is more likely (Alexandratos and Bruinsma, 2012). As
(relatively) more people will live in cities, there will be relatively fewer people working on the land to produce the
food that is necessary. Furthermore, as most of future population growth will take place in sub-Saharan Africa,
food demand will rise most rapidly in this area.
Thus, agriculture in the Global South will be fundamentally different from what it is now in less than a century.
More food will have to be produced with less people and the increasingly urban population will more and more
rely on markets to obtain the food it needs. This begs the basic question: how can we make sure that the soils
necessary to produce all this food are sustainably managed and preserved for future generations?
**Soil conservation in a changing global context**
Two contrasting pathways can be followed to meet the expected increase in food demand in the Global South.
More food can be produced either by extending the area over which current food production systems are applied
or by agricultural intensification, i.e. by increasing the amount of food produced per unit of land.
Both pathways are, in principle, possible: until present, Africa has followed the first path. Over the last five
decades, the increasing food demand of African populations has mainly been met by increasing the area used for
farming, while yields per unit of surface area remained stable and very low (Henao and Baanante, 2006). This
evolution sharply contrasts with the one observed in most parts of Asia: here agricultural production was mainly
increased through intensification (Henao and Baanante, 2006). In Asia, the Green Revolution led to a dramatic
rise in agricultural yields through the combination of new crop varieties, better farming technology and the
increased use of fertilizers. As a consequence, Asia now manages to feed its population much better than it did in
1970: the amount of available kcal per person rose from ca. 2000 kcal to ca. 2400 kcal (South Asia) or even 3000
kcal (East Asia) in 2005 (Alexandratos and Bruinsma, 2012) despite the fact that the amount of land used for
agriculture did only marginally increase (Henao and Baanante, 2006) and despite the fact that the population in
these regions increased from 0.98 billion to 1.53 billion (East Asia) and from 1.06 billion to 2.20 billion (South
Asia) over the same period (http://esa.un.org/unpd/wpp/).
While the challenge for African agriculture is not dissimilar to that of Asia in the 1960s, Africa does not necessarily
have to go down the same route. In principle, it could continue to follow the areal extension strategy policy for
some time to come. At present, ca. 290 million ha of agricultural land is in use in Africa, but another 400 million
ha of African land is suitable (good) or very suitable (prime) for agriculture (Alexandratos and Bruinsma, 2012).
Therefore, there is scope for a strategy whereby significantly more land would be used for agriculture than is the
case at present although this would pose important problems: a large fraction of the suitable land is located in
politically unstable countries and/or far from existing markets (Chamberlin et al., 2014).

An extension strategy may, at first sight, be attractive from the point of view of soil conservation. One might indeed argue that this would be based on agricultural technology that has been in use for decades, and may therefore be best suited to increase agricultural production without causing excessive soil degradation. Indeed, the occurrence of erosion in mechanised, intensive agricultural systems is often attributed to the loss of traditional soil conservation methods (Bocco, 1991). Averting intensification and aiming at area extension may therefore seem a suitable solution to avoid excessive soil degradation as traditional farming methods can be maintained and optimised to be as environmentally friendly as possible. Many organisations do indeed stress environmental protection and sustainability as key issues to be addressed in the further development of African agriculture and explicitly state that Africa should indeed follow a path different from the Asian Green Revolution (De Schutter, 2011).

While it is evident that we should learn from agricultural developments in Asia and avoid the dramatic negative effects the Asian Green Revolution had in some places, we argue here that tropical smallholder farming does need intensification for soil conservation to become successful. This intensification should be smart: it not only needs to be sustainable and to avoid jeopardising the capability of the natural resources to meet the needs of future generations. Intensification strategies should also maximise the opportunities of current and future farmers to generate an acceptable income by providing them with access to profitable markets and supplying them with the necessary knowledge and technology to produce for these markets. Smart intensification requires an approach that does not focus on the conservation of natural resources alone but also on the creation of added value using a future-oriented perspective and the quantity and quality of food production and supply. Clearly, improving the livelihood of the farmers and farming communities should be a key element. However, the capability of this farming community to provide the necessary agricultural supplies to an ever growing non-farming population also needs to be taken into account. Thus, it is not only important to consider the current socio-economic conditions but also how demographic and socio-economic conditions are likely to change in the future. We argue that smart intensification will not only make soil conservation more achievable but that it would also allow to reap additional environmental benefits that may be lost when a less intensive or less future-oriented development path is chosen. As is the case for 'smart cities', we do not believe a single, all-encompassing definition of smart intensification can be formulated. However we summarized the components that we consider to be essential in Figure 2. In the rest of the paper we focus the discussion on how soil conservation may benefit from smart intensification.

*Smart intensification will allow to spare the most erosion-prone land from agriculture thereby reducing landscape-scale erosion rates.* When farmers select land for arable production, they will select the most suitable land that is available. In general this means that, for obvious reasons, flatter land is preferred over steeper land. (Van Rompaey et al., 2001; Bakker et al., 2005). Steep lands are generally much more difficult to cultivate than flatter areas and yields can be expected to be lower in comparison to yields (for the same amount of inputs) on flat land, because soils are intrinsically less productive and/or because soil productivity is negatively affected by accelerated erosion (Stone et al., 1985; Ellis-Jones and Sims, 1995; Lu and van Ittersum, 2004). The combination of both effects (more labour required and lower yields) invariably implies that the net returns of arable farming decrease with increasing terrain steepness. The total amount of erosion as well as the amount of erosion per unit of crop yield will therefore necessarily increase when area expansion is preferred over intensification (Figure 3, Figure 5).

Increasing agricultural production in Africa through areal extension alone would therefore imply that overall soil
losses would increase much more rapidly than agricultural production would. If, on the other hand agricultural
yields on good agricultural land would be improved, it may be possible to set aside some of the marginal land that
is currently used for arable farming. The somewhat counterintuitive result of this will be that, even if erosion rates
on the arable land that remains in production would increase due to intensification, the overall soil loss (at the
landscape scale) would still decrease (Figure 3).
*Smart intensification will conserve soil carbon which will, on its turn, reduce erosion risks.* Over the last decades,
a significant body of scientific literature has emerged on the potential of agricultural land to store additional soil
organic carbon through the use of appropriate management techniques. While studies do suggest that some gains
are indeed possible, most studies report modest gains at best. Reported average sequestration rates under
conservation tillage in Canada are between 0 and 0.14 t C ha$^{-1}$ yr$^{-1}$ in Canada (VandenBygaart et al., 2010) while
an average sequestration rate of 0.12 t C ha$^{-1}$ yr$^{-1}$ has been calculated for the USA (Eagle et al., 2012). In a study
covering 12 study sites in three Midwestern states of the USA Christopher et al. (2009) did not find any significant
increase in soil organic carbon storage under no-till in real farming conditions. Experimental studies also showed
that under agroforestry gains in soil organic carbon are small, with an average of 0.25 t C ha$^{-1}$ yr$^{-1}$ (Govers et al.,
2013). These findings contrast not only with claims in the literature (Ramachandran Nair et al., 2009), but also
with the observation that soil carbon stocks on natural (or undisturbed) land are generally much higher (often more
than three times higher) than those observed on arable land (e.g. Poeplau et al., 2011; Hiederer and Köchyl, 2012).
The latter is related to two main factors: (i) biomass is not removed from natural land, which results in larger
organic carbon inputs and (ii) these lands are not mechanically disturbed which reduces carbon respiration rates.
Thus, more soil carbon will be conserved when the extent of agricultural land is reduced and more land is preserved
under or restored towards natural conditions. An additional beneficial effect of the latter is that soil organic carbon
stocks may increase on agricultural land with increasing agricultural yields, provided that the residual biomass is
adequately managed (VandenBygaart et al., 2010; Minasny et al., 2012): this, in turn, will reduce the erosion and
degradation risk (Torri and Poesen, 1997). Thus, intensification will allow to preserve more carbon then areal
extension (Figure 3, Figure 5).The fact that intensification is beneficial for soil carbon conservation has also been
demonstrated at the global level: agricultural intensification has allowed to avoid ca. 161 Gt of carbon emissions
from the soil to the atmosphere between 1960 and 2005 (Burney et al., 2010).
*Smart intensification will help to make agriculture in the South more water-efficient.* Agriculture is by far the
largest global consumptive user of blue water (water extracted from rivers and groundwater): at the global scale,
over 80% of all consumptive water use is related to agricultural activities (e.g. Doll et al., 2009). As the amount
of available water will not significantly increase in the future, a more efficient water use is a prerequisite to increase
agricultural production in the South. Less productive systems are often more water-intensive, i.e. more units of
water are needed for each unit of crop that is produced. Striving towards higher yields will remedy this problem
as it allows to increase the amount of crop produced per unit of water (Rockström et al., 2007). Higher yields are
therefore a means to increase water conservation and to make sure that more water is available for the functioning
of non-agricultural ecosystems. Clearly, the realisation of this potential requires other measures as well such as a
realistic pricing of water and water use monitoring in areas where water scarcity is a problem so that inefficient
use of this scarce resource can be prevented. Again, the implementation of such systems will be far more efficient
in high-yield systems as the return per unit of capital cost will be higher.
*Smart intensification is beneficial for biodiversity at the landscape scale.* Environments where intensive
agriculture is dominant are often very poor in terms of biodiversity. One might therefore suggest that, in order to
preserve biodiversity, one should avoid intensification and maintain a certain biodiversity on agricultural lands.
Again, such a strategy would necessarily imply that more land would be needed to produce the same amount of
agricultural goods. Recent studies have consistently shown that such a strategy is not beneficial for biodiversity at
a larger scale: the biodiversity gained on agricultural land is, in general, not sufficient to compensate for the
additional biodiversity loss due to agricultural land expansion (e.g. Phalan et al., 2011b; De Beenhouwer et al.,
2013; Schneider et al., 2014). Thus, land sparing and concentrating intensive agriculture on designated areas is
generally a better strategy than land sharing with low-intensity agriculture that will occupy a much larger fraction
of the available land (Figure 5). Sparing will not always be the best strategy as this will depend on local conditions:
for instance, wildlife-friendly agriculture may be the best solution in the buffer zones around wildlife reserves.
*Smart intensification will increase the added value of the land used for agricultural production and hence make*
*the implementation of conservation measures economically sound.* Clearly the economic value of a good such as
arable land depends on the economic return that can be gained from the use of it. Intensification will allow to
increase these returns. This is especially true for sub-Saharan Africa where yields are still abysmally low
(Neumann et al., 2010). While there are many reasons for this, a key factor is that African soils are chronically
underfertilized (Henao and Baanante, 2006; Keating et al., 2010). The amount of fertilizer used per unit of surface
are of agricultural land in Africa is only 10% of what is being used in Europe or the United States: the consequence
is that, in many cases, the nutrient balance of many African agricultural systems is negative, i.e. more nutrients are
removed through harvesting than there are supplied by fertilization (Smaling et al., 1993; Henao and Baanante,
2006). This negative balance is further aggravated by soil erosion, which annually mobilises more nutrients than
are applied in sub-Saharan Africa (Quinton et al., 2010). Even a modest increase in fertilizer use may therefore
allow to significantly boost agricultural yields in sub-Saharan Africa, at least if this increase would be accompanied
by other measures such as the introduction of high-yield varieties and the necessary training for the farmers
(Sanchez, 2010; Twomlow et al., 2010; Mueller et al., 2012).
Higher agricultural yields will increase the added value that may be produced per unit of agricultural land and
hence its value. A consequence of this is that the economic stimulus to implement conservation measures on this
land will increase as land will become a more precious resource. Furthermore, intensification will also reduce the
overall conservation investment that has to be made as the acreage that needs to be treated will be smaller which
will allow to concentrate the available resources on a smaller area. Finally, many conservation strategies are based
on the use of crop residue (i) to return  nutrients and carbon to the soil and (ii) to reduce the soil erosion risk. Such
strategies are likely to be more successful when more residue per unit of area is available. Case studies have
repeatedly shown that the mechanisms described above can indeed lead to more effective soil conservation under
increasing intensification and population pressure  (e.g. Tiffen et al., 1994; Boyd and Slaymaker, 2000)
*Smart intensification will help to create the market opportunities needed for sustainable agriculture.* The dramatic
increase in population that will occur in the South over the next century, in combination with rapid urbanisation
and economic growth, make the transition towards a market-oriented agriculture inevitable. This is not a bad thing:

all too often we have a far too rosy view on the potential of subsistence agriculture. The truth is that subsistence farming does not generate the necessary financial means for the farmers to get out of poverty, although improvements in agricultural technology may contribute to increased food security (Harris and Orr, 2014). Only when farmers have access to markets they can generate an income that allows them to fully participate in society so that they can not only benefit from the material perks of modern life but also provide a high quality education to their children and the necessary health care to those who need it: soil conservation as such cannot achieve this (Posthumus and Stroosnijder, 2010). Case studies support that a symbiosis between the development of a market-oriented agriculture and soil conservation is indeed likely as market access provides farmers with the economic incentives to implement soil conservation measures (Boyd and Slaymaker, 2000). Again, the transition from a subsistence to a market-oriented system will almost inevitably have to be accompanied by intensification as the latter will allow a better return on both capital and input investment.

*Smart intensification will not be sufficient to achieve adequate soil conservation (but it will help).* The points raised above illustrate that adequate soil conservation is much more likely to be achieved if more intensive agricultural systems are developed in the Global South as the economic and environmental stimuli to implement soil conservation measures will be much larger. Yet, the experiences in Europe and Northern America illustrate that this may not be sufficient to achieve adequate soil conservation and that government stimulation (through financial measures) and/or coercion may be necessary to further reduce soil degradation. It is, however, the magnitude of such efforts and their effectiveness that should be considered. The societal efforts and costs that will be needed to achieve adequate soil conservation will be far smaller when less land is used for agriculture as much less land will need treatment. Furthermore, one may also imagine that efforts to convince farmers to adopt conservation measures will be more successful in an intensive, market-oriented agricultural system as they will, generally, be more open to changes and both governments and other stakeholders will have more leverage in discussions on how the agricultural system needs to be organised. This is, obviously, no guarantee for success as potential direct financial benefits may seduce the stakeholders to neglect the necessary investments to achieve long-term sustainability. The latter is a problem that occurs everywhere where environmental and economic concerns conflict and, while general principles to resolve such problems have been formulated (Ostrom, 2009), specific policies to deal with this conflict will depend on local conditions.

**Conclusions**

All too often, soil conservation is discussed in isolation, whereby much attention is given to the effectiveness of technical solutions in reducing excessive soil and water losses at a given location. Agriculture, however, is a system wherein lateral connections at different scales are very important: actions at a specific location will necessarily have implications at other locations. Agricultural systems are also subject to constant change as they respond to changes in population numbers, population distribution, economic wealth and cultural preferences. A coherent vision on the development of soil conservation in 21$^{st}$ century needs to account for this context and needs to consider both the spatial and temporal dynamics of agricultural systems.

While it is certainly true that conservation technology can be further developed other considerations may be more important for the successful implementation of soil conservation programs. In our view, smart intensification is an essential ingredient of any strategy seeking efficient soil conservation while at the same time meeting the growing

food demands of a strongly increasing, more urbanised global population. Smart intensification will help to reduce
the land surface area exposed to a high soil degradation risk while it will, at the same time, increase the return on
the soil conservation measures that will still be necessary. Smart intensification will also allow to reap additional
environmental benefits in terms of soil organic carbon storage, biodiversity and water availability. It will also be
directly beneficial to the farmer, allowing her/him to produce food for more people and to achieve an acceptable
income. It is therefore no surprise that, when considering these other angles, other researchers have reached similar
conclusions, stating that agriculture in the Global South and particularly in Africa needs to intensify and that the
exclusive focus on smallholders as engines for growth needs to change (Collier and Dercon, 2009).
Intensification is not a panacea that magically solves all problems. Striving towards higher crop yields will require
the use of more external inputs, including the use of mineral fertilizers. This is often assumed to be detrimental to
the environment: yet this only will be true if fertilizers are used excessively, as is the case now in many areas of
the world (Sattari et al., 2012; Lassaletta et al., 2014; Zhang et al., 2015). If correctly used, the environmental
benefits of judicious mineral fertilizer use will more often than not outweigh their potential negative impacts by
reducing the amount of land needed for agricultural production (Tilman et al., 2011). Furthermore, intensification
will require higher energy and capital inputs per unit of surface area: these extra investments will partly be
compensated by the fact that a smaller area of land needs to be cultivated but access to markets will often be
essential to make intensification profitable.
Smart intensification as such will not be sufficient to reduce soil loss to acceptable levels: also in intensive systems,
soil losses are often higher than is tolerable and conflicts between (long-term) environmental and (short-term)
economic goals will be present. Yet, they will be easier to tackle when we give smart intensification adequate
consideration in any plan on future agricultural development in the Global South.
**Acknowledgement**
This paper greatly benefited from the critical comments of Martin Van Ittersum and an anonymous referee. The
financial support of STAP (Scientific and Technical Advisory Panel of the Global Environmental Facility) and the
IUAP project SOGLO (The Soil System under Global Change, IUAP P7/24) is gratefully acknowledged.

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

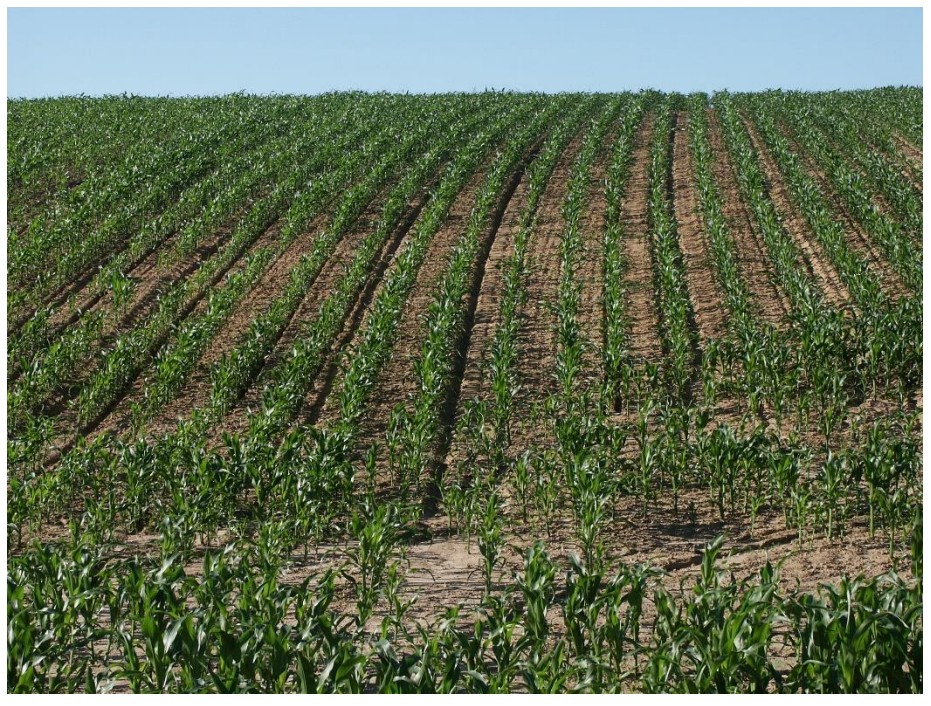


**Figure 1. The presence of a dense network of rills and of significant deposition at the footslope (here in Huldenberg,**
**Belgium in July 2006) is a such sufficient proof for excessive soil erosion (in this case erosion exceeded 100 t ha$^{-1}$ in a**
**single event)**

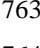

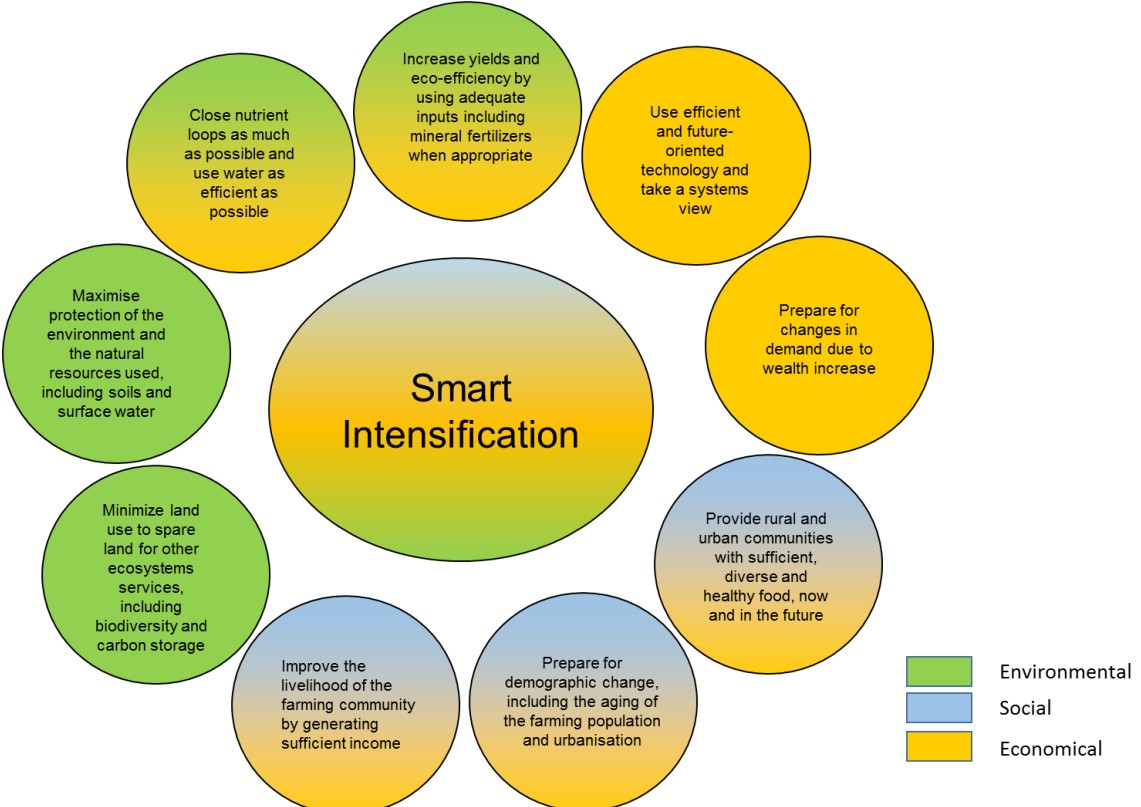

**Figure 2 Different aspects of smart agricultural intensification. Colouring refers to main reason as to why each aspect is important.**

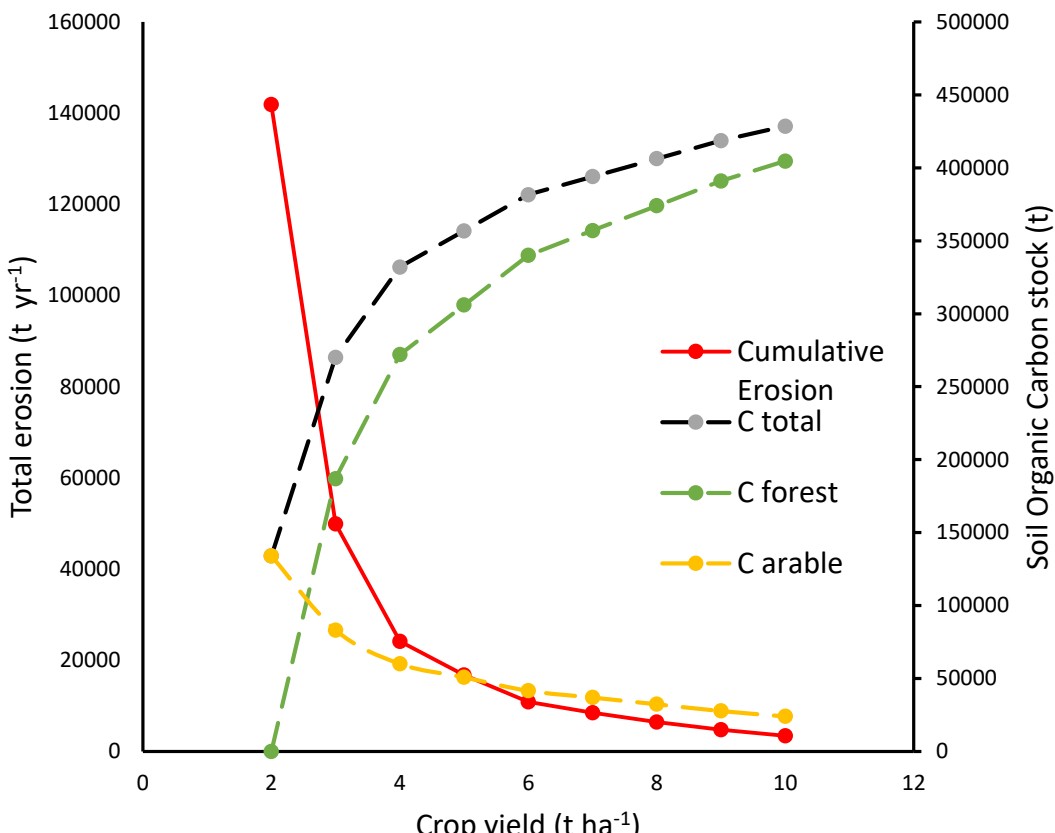

770

**Figure 3 Modelled total erosion (t, yr⁻¹ left axis) and soil organic carbon stocks (t, right axis) vs. crop yield per ha for a hypothetical test area of 2900 ha and assuming a total cereal production of 5000 ton. We assumed that slope gradients (sin θ) were uniformly distributed between 0.02 and 0.58, i.e. an area of 100 ha in each 0.02 slope class. The crop yield shown is the crop yield on a zero slope and relative crop yield ($P$) is assumed to vary with slope: $P=1-(\sin\theta)^{0.5}$. Erosion ($E$, t ha⁻¹ yr⁻¹) is assumed to vary with slope gradient according to the slope function derived by Nearing (1997): $E \sim -1.5+17/[1+\exp(2.3-6.1\sin\theta)]$, and an erosion rate of 10 t ha⁻¹ yr⁻¹ is assumed on a 0.09 slope. Soil organic carbon stocks per unit area are assumed to be 40 t ha⁻¹ on arable land and 170 t ha⁻¹ under forest (Poeplau et al., 2011). The total soil organic carbon stock (C total) in the area strongly increases with increasing crop yield because the gain in soil organic carbon stocks on forested land (C forest) is much more important than the loss on arable land (C arable).**




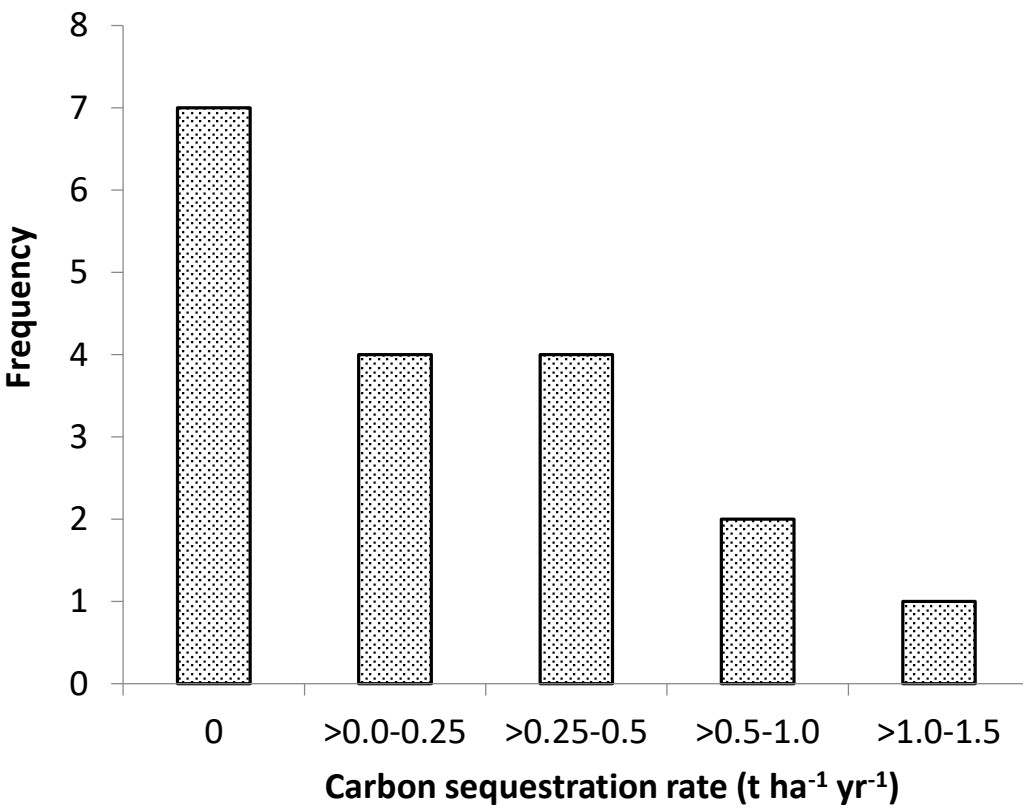


**Figure 4 Frequency distribution of experimentally observed carbon sequestration rates under agroforestry. Data from 18 paired field studies in both (sub-)tropical and temperate climates (details and references of studies in (Govers et al., 2013)). The average soil organic carbon sequestration rate reported over all 18 studies is 0.25±0.33 t ha$^{-1}$ yr$^{-1}$.**


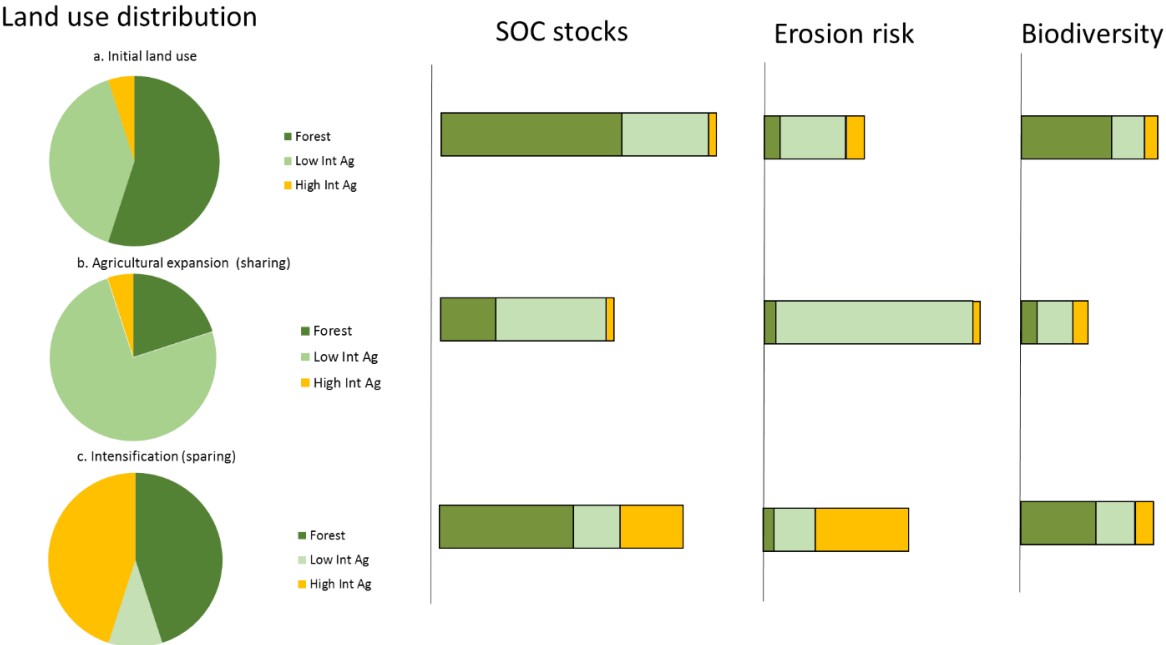


**Figure 5 Semi-quantitative illustration of the effects of a significant increase of agricultural production through smart intensification (sparing land) vs. agricultural expansion (sharing land) on soil organic carbon stocks, the erosion risk and biodiversity. We assume that in a given area the required increase in agricultural production is such that, if yields are not increased, the entire area that is potentially suitable for agriculture (80% of the total area) has to be used for agriculture and that smart intensification would reduce the area needed to ca. 55% of the total area. The bar graphs give a semi-quantitative assessment, at the landscape scale, of the impact of these alternatives according to current scientific insights. Smart intensification is beneficial with respect to soil organic carbon storage because soil organic stocks under natural forest are much higher than under arable land (e.g. Poeplau et al., 2011) . Smart intensification will reduce total soil erosion because less marginal (sloping) land needs to be taken into production (e.g. Van Rompaey et al., 2002). Finally, smart intensification is beneficial for biodiversity because more forest is preserved and the biodiversity of undisturbed forests is much higher than that of land used for agriculture (e.g. Phalan et al., 2011a).**



| Region | Condition | Trend |
|---|---|---|
| Asia | Poor | Negative |
| Latin America | Poor | Negative |
| Near East and North Africa | Very Poor | Negative |
| Sub-Saharan Africa | Poor | Negative |
| Europe and Eurasia | Fair | Positive |
| Northern America | Fair | Positive |
| Southwest Pacific | Fair | Positive |

**Table 1 Conditions and trends with respect to soil erosion as assessed by experts (data from FAO and ITPS, 2015)**