# Peer review of "Soil Conservation in the 21st Century: Why we need Smart 5 Agricultural Intensification"

_SOIL, 2016_

## Referee Comment (RC1) · Anonymous Referee #1 · 26 Sep 2016

At first I want to congratulate the authors for putting together such an important piece of work. Although the manuscript is quite long (particularly the Introduction and "The status of soil conservation") it was a pleasure for me to read it. It gives a well written summary of the problem we have to solve and a vision how to further proceed. It addresses often ignored relationships between natural and socio-economic processes. I have just some minor suggestions to further improve this manuscript:

I would highly appreciate to have a summarizing figure about the different aspects of smart intensification as described in the manuscript including the most important measures of such a smart intensification. Particularly the latter is weakly developed in the whole paper. Is there any option to include organic management in such a strategy of smart intensification?

[Figure]

I like Figure 1 very much to illustrate the benefits of smart intensification. However, I miss the quantitative base of this figure – where do the numbers come from?

Table 2 summarizes very well the C sequestration rates under agroforestry. If these numbers have been already published in the cited reference (please check the list – it is mentioned 2 times) I would not spend 1,5 pages to support one sentence in the manuscript. Instead, I would highly appreciate an overview of erosion rates, rates of soil formation and tolerable levels of soil erosion including a brief discussion about the concept of tolerable levels of soil erosion.

The paper might profit from a short explanation of the contradiction between the severe soil degradation on the one hand and almost no direct economic benefit from soil conservation on the other hand.

Some detailed comments: Abstract: Please add some details about the concept of smart intensification and some reasons for the predicted positive effects. Line 137: Please check Line 486, line 512: Dot is missing

---

## Referee Comment (RC2) · M. van Ittersum (Referee) · 7 Oct 2016

This is a very well written manuscript that makes a plea for a smart intensification of agriculture to allow for effective soil conservation in the 21st century. One may argue it comes close to an opinion paper rather than a research of review paper, but it is consistent in its reasoning and it is generally well backed up by references and some data. I largely agree with the reasoning of the authors and I think it is well worth to publish such a message. But clearly the limitation is that the piece is largely qualitative as is illustrated by Table 1 and Figure 1. I think the authors should emphasize that Figure 1 is illustrative and not based on any data – if that is not true the authors must indicate the basis for their 'quantities'. Also, some kind of definition of Smart intensification is needed: do we really need such term, or would sustainable intensification be equally good? Please no more confusing terms than strictly necessary!

[Figure]

Other than that, I have no reservation to recommend the manuscript for publication after minor revisions:

1. I suggest to add to the title 'of agriculture' and consider to substitute smart by sustainable 2. The abstract is rather vague and general. It would be good to make it more specific in its conclusions. 3. Line 45: important => sizeable 4. Line 90: quantities per ha may be more meaningsful for readers. 5. Line 101: delete 'is' 6. Line 103: phosphorous => phosphorus 7. Line 111: takes => take 8. Line 112: ITPS in full please 9. Line 128: delete 'different' 10. Line 137: is needed => data? 11. Line 146: give examples of such spatial data please 12. Line 153-165: Some elaboration on lateral spatial effects is justified. 13. Line 180: delete because 14. Line 257-260: unclear sentence; please reprhase 15. Line 284-6: see more recent reviews on conservation agriculture and tillage, e.g. by Van Kessel and others 16. Line 330: These => This 17. Line 333-5: unclear sentence; please rephrase 18. Line 371-374: see e.g. Van Vliet et al., 2015 in Global Food Security 19. Line 387-9: FAO projects 60-70% which is probably closer to the real number. 20. Line 390: less => fewer 21. Line 421-2: please give numbers in just two digits 22. Line 429: of => of 23. Line 427-430: please check Chamberlin et al. (2014) in Food Security: clearly there are indications there is less land 'available' in SSA. 24. Line 450: call => called? 25. Line 486: full stop after the reference. 26. Line 506: delete indeed 27. Line 513: inventories => storage? 28. Line 519: inventories??? 29. Line 523: preserver => preserve 30. Line 556: is problem => is a problem 31. Line 572: delete be 32. Line 602: location => location 33. Line 609: While it most => While it is most 34. Line 609: insert comma after developed 35. Line 618: people??? 36. Line 625: I do not think Foley et al. (2011) is an appropriate reference for this statement. I would rather use Zhang et al. (2015) in Nature, Lasseletta et al. in Env. Res. Letters (2014) and Sattari et al. (2012) in PNAS. 37. Line 629: Delete Clearly 38. Line 631: economical => economic 39. Line 632: easier => be easier 40. Line 637: storsks?? Stocks perhaps? 41. Figure 1: please clearly indicate this figure is just illustrative but not meant to be quantitative. If this is not the case, then please back it up with data! 42. Table 1: it would be good to
define poor, fair, negative, etc.

---

## Editor Comment (EC1) · J. Wallinga (Editor) · 10 Oct 2016

Dear Gerard,

As you've seen two reviewers have now commented on your manuscript 'soil conservation in the 21st century: why we need smart intensification'.

Both referees highly value your manuscript and support publication in the special Issue of SOIL, provided that some improvements / corrections / justifications are made.

I fully agree with the referees recommendation; the paper is convincing and well backed up by literature, and reads very smoothly. The modifications suggested by the referees will further strengthen the publication, and I would like to ask you to consider the suggestions, and make the modifications where you think appropriate. Please provide a

detailed rebuttal where you decide not to follow the referees' suggestions.

Seeing the relatively limited revisions required, I would like to ask you to submit the revised manuscript by the end of October. Please let me know whether this is feasible.

Best regards, Jakob Wallinga co-editor special issue
* * *

---

## Author Comment (AC1) · 4 Dec 2016

Dear Editors,

Attached you will find my replies to the referees as well as a revised version of my manuscript. Unfortunately I was stupid enought to not switch on 'track changes' when revising my manuscript. However, in the referee comments you will find annotations as to how nearly all referee comments were taken into account. The manuscript has also been improved on the following points:

- A signficant number of references has been added to support some of the points made in the manuscript - Three figures were added, one is a photograph simply illustrating the point that excessive erosion can often be readily detected and one which quantitatively supports the point that intensification may indeed help to store

more carbon and reduce erosion at the landscape scale. - Language was revised throughout the manuscript - Bibliography revised and updated

Please also note the supplement to this comment:
http://www.soil-discuss.net/soil-2016-36/soil-2016-36-AC1-supplement.zip

---

## Author Response (AR1)

Reply to the Editorial remarks:

Beste Jakob en Saskia,

Ik heb de laatste suggesties van Jakob nu opgenomen in de paper.  Ik denk dat ik ze allemaal heb behandeld en ben ook wel erg tevreden met het resulaat ;-). Ik kan op dit moment de paper niet uploaden in het Copernicus-systeem maar hij is alvast aangehecht, zowel in Word met track changes als in pdf.  Als Jakob nog eens toestemming kan geven, dan laad ik ook alles op in het Copernicus-systeem.

Vele groeten, Gerard

-----Original Message-----
From: Wallinga, Jakob [mailto:jakob.wallinga@wur.nl]
Sent: 20 December 2016 21:41
To: Gerard Govers <gerard.govers@kuleuven.be>
Cc: Keesstra, Saskia <saskia.keesstra@wur.nl>
Subject: Re: paper SOIL

Beste Gerard,

Nogmaals dank voor deze mooie bijdrage aan het special issue.

Zoals beloofd nog enkele opmerkingen en suggesties. De versie die ik bekeken heb is de versie die je eerder per mail gestuurd had (en de regel nummers daarin). Vetgedrukt woorden zijn suggesties voor aanvullingen, italics geeft een aanpassing weer.  Ik zag net dat de versie die geupload is nog iets afwijkt van de mail versie (oa caption fig. 4 en mogelijk meer); in dat geval zijn enkele opmerkingen mogelijk niet meer relevant.

Ik heb zonet ook een officiele respons in het editorial systeem zetten, opdat je de finale versie kunt uploaden.

Algemeen:
waar meerdere referenties worden gegeven: spatie ontbreekt na de ;

Comments ref 1:
I would highly appreciate to have a summarizing figure about the different aspects of smart intensification as described in the manuscript including the most important measures of such a smart intensification. Particularly the latter is weakly developed in the whole paper. Is there any option to include organic management in such a strategy of smart intensification?

Ik kon niet goed vinden welke aanpassingen gedaan zijn in het MS in response op deze suggestie; zou je kunnen aangeven welke aanpassingen gedaan zijn, of uit kunnen leggen waarom deze suggestie niet is gevolgd?

*We hebben nu een figuur toegevoegd die naar onze mening de voornaamste elementen van smart intensification samenvat. We gaan bewust niet erg diep in op het verduidelijken van al die concepten omdat dat de paper veel te lang zou maken. Wat we wel duidelijk hebben proberen te maken (met een kleurcode) is dat het niet enkel gaat om environmental protection, noch enkel om economie. Het gaat, naar onze mening, om intensificatiestrategiën die een veelvoud van criteria in acht nemen die toekomstgericht, eco-efficiënt en sociaal acceptabel zijn.*

Overige opmerkingen

Regel:

87- ANUAL amount of  fertilizers

*Changed*

billion US per year

*Changed to US $*

- and Asia (ADD REF).

*Ref added*

- economic activities,

*Changed*

- missende spatie na vegetation

*Changed*

- is not always be

*Changed*

- ridges, and/or

*Changed*

- as an important

*Changed*

- moving there as an opportunity

*Changed*

- 298 - gehele zin checken en verbeteren

*Changed*

- amount of available kcal PER PERSON?

*Changed*

- Foregoing vervangen door Averting of door Preventing

*Changed*

- that it also allow > checken en verbeteren

*Changed*

- verwijzing naar Fig. 4 weglaten, of volgorde fig. 3 en 4 wijzigen

*Order changed*

Figure 2:
Deze figuur wordt duidelijker als de legenda in het figuur wordt geplaatst, en de afkortingen in de legenda worden vervangen door Cumulative erosion; C total; C arable land; C forest. Pijlen welke as (links of rechts) van toepassing werkt verduidelijkend.

Rephrase caption fig. 2:
Modelled cumulative erosion and carbon stocks as a function of crop yield in a hypothetical test area of 2900 ha for a total cerial yelad of 5000 ton. The total carbon stock (C total) is the combined carbon stock of arable land (C arable land) and that of forest (C forest)
*We rephrased the whole caption and think that it reads a lot better now*

- Give the s lope function of Nearing
*Added*

- are assumed to be
*Changed*

- refs needed for statements:  gains are indeed possible (e.g. REFS), most report modest gains at best (e.g. REFS)
*This sentence summarizes what follows below: we have extended the number of references in this section and referred the readers to various studies confirming this general statement.*

- not clear whether 0.12 in USA or between 0 and 0.12.... > clarify

*Clarfied*

- Christopher et al. (2009)

*Changed*

- error - reference not found0

*Changed*

Figure 3:

labels are confusing: the second bar is for 0.0 to 0.25, so suggests to include 0 (which is given in the
first bar......)
*Changedd*
388- carbon sequestration rates in arable land?
389 - Govers et al. (2013)
*Both changed*
Figure 4:
This figure needs more context / explanation. What is it based on (expert judgment, model,
literature?). The pie charts and bars suggest at least semi-quantitative information, even if it is based
on estimates or guesstimates; qualitative information cannot be presented in this form. In addition,
in the main text, it is suggested that intensification will/may allow a reduction of the Agricultural
land; whereas this figure suggests that intensification will still result in a decrease of natural (forest)
land. Please make sure that the figure is in line with the text, or explain the differences.
*We have rewritten the whole caption of the figure to make it clearer what we mean. Indeed, the*
*information is semi-quantative rather than qualitative and this has been chaned.*
420 – therefore
*Changed*
429 - reserves > add ref
*I did look for such a refrerence but did not find it, so I changed the statement to 'may be'*
452 - (e.g. Tiffen et al., 19994; Boyd ...    (if statement repeatedly shown is correct)
*Statement is correct: Boyd and Slaymaker refer to several studies, but I have added e.g. anyway ;-)*

Met hartelijke groet,
jakob
----------------------------------------------------------------------------
Prof. Dr. Jakob Wallinga
Soil Geography and Landscape group, WUR (group leader)
Netherlands Centre for Luminescence dating (director)
Droevendaalsesteeg 3 | 6708 PB Wageningen (GAIA, room B.123)
+31(0)317 484040 | PO Box 47 | 6700 AA Wageningen | Netherlands
----------------------------------------------------------------------------
________________________________

From: Gerard Govers <gerard.govers@kuleuven.be>
Sent: 19 December 2016 13:26
To: Wallinga, Jakob; Keesstra, Saskia
Subject: RE: paper SOIL

OK, prima, ik kijk er naar uit !

Vele groeten, Gerard

From: Wallinga, Jakob [mailto:jakob.wallinga@wur.nl]
Sent: 19 December 2016 09:29
To: Keesstra, Saskia <saskia.keesstra@wur.nl>; Gerard Govers <gerard.govers@kuleuven.be>
Subject: RE: paper SOIL

Klopt Saskia,

@Gerard: ik ben jullie paper nogmaals met veel plezier aan het doorlezen. Complimenten voor het artikel, dat leest als een wetenschappelijk onderbouwt pamflet. Een zeer goede bijdrage voor ons special issue.

Ik zal je vandaag of uiterlijk morgen een mail sturen met kleine correcties die nog doorgevoerd moeten worden (enkele zinnen die niet lopen).

Met hartelijke groet,
Jakob
* * *
Prof. Dr. Jakob Wallinga
Soil Geography and Landscape group, Wageningen University (group leader) Netherlands Centre for Luminescence dating (director) Droevendaalsesteeg 3 | 6708 PB Wageningen (GAIA building, room B123)
+31 (0)317 484040 | PO Box 47 | 6700 AA Wageningen | The Netherlands
* * *
From: Keesstra, Saskia
Sent: Monday, December 19, 2016 9:26 AM
To: Gerard Govers; Wallinga, Jakob
Subject: RE: paper SOIL

Beste Gerard
Jakob moet nu je paper beoordelen en dan goedkeuren (neem ik aan). Daarna kan het worden gepubliceerd.
Groetjes
Saskia

From: Gerard Govers [mailto:gerard.govers@kuleuven.be]
Sent: zaterdag 17 december 2016 5:13
To: Wallinga, Jakob; Keesstra, Saskia
Subject: paper SOIL

Dag Jakob, Saskia,

Is er nog iets wat ik moet doen voor de SOIL paper ? Ik denk dat ik alles heb opgeladen: laat me
weten als er nog iets moet gebeuren.

Vele groeten, Gerard

Gerard Govers
Department of Earth and Environmental Sciences KU Leuven<http://ees.kuleuven.be/> Director of
Arenberg Doctoral School<http://set.kuleuven.be/phd>
Discover how you shape the world by making a PhD at the most innovative university of the
European continent<http://www.shapetheworld.eu/>

Field Code Changed

Field Code Changed

Field Code Changed

[revised manuscript text omitted]

Field Code Changed
Field Code Changed
Field Code Changed
Field Code Changed
Field Code Changed

[Figure]

[Figure]

**Figure 32 Modelled Ccumulative erosion (Ecum) (left axis) and soil organic carbon stocks (right axis) vs. crop yield per ha for a hypothetical test area of 2900 ha and assuming a total cereal production of 5000 ton.  We assumed that slope gradients (S (tan)) were uniformly distributed between 0.02 and 0.58, i.e. an area of 100 ha in each 0.02 slope class. The crop yield shown is the crop yield on a zero slope and relative crop yield (P) is assumed to vary with slope (P=1-S$^{0.5}$). Erosion is assumed to vary with slope gradient according to the slope function derived by Nearing (1997) and an erosion rate of 10t ha$^{-1}$ y$^{-1}$ is assumed on a 0.09 slope. Soil organic carbon stocks per unit area are assumed to be 40 Mg ha$^{-1}$ on arable land and 170 Mg ha$^{-1}$ under forest (Poeplau et al., 2011). The total soil organic carbon stock (C total) clearly increases with increasing crop yield because the gain in soil organic carbon stocks on forested land (C forest) is much more important than the loss on arable land (C arable). **

Increasing agricultural production in Africa through areal extension alone would therefore imply that overall soil losses would increase much more rapidly than agricultural production would. If, on the other hand agricultural yields on good agricultural land would be improved, it may be possible to set aside some of the marginal land that is currently used for arable farming. The somewhat counterintuitive result of this will be that, even if erosion rates on the arable land that remains in production would increase due to intensification, the overall soil loss (at the landscape scale) would still decrease (Figure 3Figure 2).

*Smart intensification will conserve soil carbon which will, on its turn, reduce erosion risks.* Over the last decades, a significant body of scientific literature has emerged on the potential of agricultural land to store additional soil organic carbon through the use of appropriate management techniques. While studies do suggest that some gains are indeed possible, most studies report modest gains at best.  Reported average sequestration rates under conservation tillage in Canada are between 0 and 0.14 Mg C ha$^{-1}$ yr$^{-1}$ in Canada (VandenBygaart et al., 2010) and while an average sequestration rate of 0.12 Mg C ha$^{-1}$ yr$^{-1}$ in has been calculated for the USA (Eagle et al., 2012). In a study covering 12 study sites in three Midwestern states of the USA

Christopher et al. (2009) did not find any significant increase in soil organic carbon storage under no-till in real farming conditions. Experimental studies also showed that under agroforestry gains in soil organic carbon are very small, with an average of 0.25 Mg C ha$^{-1}$ yr$^{-1}$ (Govers et al., 2013) (Error! Reference source not found.). 
[revised manuscript text omitted]

13, 4735-4750, 2016.